# Downscaling system for modelling of atmospheric composition on regional, urban and street scales

Roman Nuterman[1], Alexander Mahura[2], Alexander Baklanov[3,1], Bjarne Amstrup[4], Ashraf Zakey[5]

[1]Niels Bohr Institute, University of Copenhagen, Copenhagen, DK-2100, Denmark
[2]Institute for Atmospheric and Earth System Research, University of Helsinki, Helsinki, 00560, Finland
[3]World Meteorological Organization, Geneva 2, CH-1211, Switzerland
[4]Danish Meteorological Institute, Copenhagen, DK-2100, Denmark
[5]Egyptian Meteorological Authority, Cairo, 11784, Egypt

*Correspondence to*: Roman Nuterman (nuterman@nbi.ku.dk)

**Abstract.** In this study, the downscaling modelling chain for prediction of weather and atmospheric composition is described and evaluated against observations. The chain consists of interfaced models for forecasting at different spatio-temporal scales and run in a semi-operational mode. The forecasts were performed for European (EU) regional and Danish (DK) sub-regional/urban-scales by the offline coupled numerical weather prediction HIRLAM and atmospheric chemical transport CAMx models, and for Copenhagen city/street scale - by the online coupled computational fluid dynamics M2UE model.

The results showed elevated $NO_x$ and lowered $O_3$ concentrations over major urban, industrial and transport land/water routes in both the EU and DK domain forecasts. The $O_3$ diurnal cycle predictions in both these domains were equally good, although $O_3$ values were closer to observations for Denmark. At the same time, the DK forecast of $NO_x$ levels was more biased (with better prediction score of the diurnal cycle) than EU forecast, indicating a necessity to adjust emission rates. Further downscaling to the street level (Copenhagen) indicated that the $NO_x$ pollution was 2-fold higher on weekend and

more than 5 times higher during working day with high pollution episode. Despite of high uncertainty in road traffic emissions, the street-scale model captured well the $NO_x$ diurnal cycle and onset of elevated pollution episode.

The demonstrated downscaling system could be used in future online integrated meteorology and air quality research and operational forecasting, as well as applied for impact assessments on environment, population and decision making for emergency preparedness and safety measures planning.

## 1 Introduction

A progress in numerical weather and atmospheric composition modelling and continuously increasing supercomputing power make it possible to perform downscaling and nesting from the global to the local scales reaching necessary horizontal and vertical resolutions for a very detailed local meteorology and pollution forecasts. The global and regional scale Atmospheric Chemistry Transport (ACT) modelling systems actively developed and applied during the last decade. These

were developed and applied within the frameworks of multiple EU FP6, FP7 and H2020 Monitoring of Atmospheric

Composition and Climate (MACC) projects (Hollingsworth et al., 2008; Simmons, 2010; Peuch et al., 2014; Peuch et al., 2016). Such systems simulate mostly the regional background of air pollution, for instance, in the Copernicus Atmosphere Monitoring Service (CAMS; Grasso, 2017). These large-scale models with increased resolution have limited performance improvement for rural areas (Schaap et al., 2015), but have better modelling scores at reproducing spatio-temporal pollution

gradients in and near large urban areas. As a matter of fact, urban and street air pollution is usually several times higher than the pollution at regional scale over suburbs and rural areas and is mostly associated with both local emission sources (Falasca and Curci, 2018) and dominating urban weather conditions (González et al., 2018). Therefore, improvements in performance of high-resolution models for urban areas are closely linked to availability and quality of emission inventories at high spatio-temporal resolution as well as accurate simulation of urban meteorology (Mahura et al., 2008a; Mahura et al.,

2008b; Pepe et al., 2016).

Generally, within the deterministic modelling approach, there are two ways applicable for downscaling modelling. It can be performed with a single model capable of local grid refinement (Gettelman et al., 2018) or nesting of numerical grids (Petersen et al., 2005; Lauwaet et al., 2013). It can be also realised as a chain of different resolution models interfacing with each other (Temel et al., 2018; Veratti et al., 2020). In the latter case, the need of different models is justified by the fact that

mesoscale models cannot capture local scale processes accurately since the fine-scale flow patterns should be explicitly resolved rather than parameterized (Schlünzen et al., 2011; Ching, 2013). As for the air-quality prediction systems, an offline (Kukkonen et al., 2012) and online (Baklanov et al., 2014) approaches of coupling meteorological and chemical transport models have been commonly used. Each of these has its own advantages and limitations.

The goal of this paper is to demonstrate and evaluate applicability of the downscaling system coupled with computational

fluid dynamics model on an example of operational modelling of an acute air-pollution episode. This system consists of several models with nested numerical grids where meteorological and atmospheric composition prediction on regional- and sub-regional/urban-scales is performed. Three interfaced and coupled models – (i) numerical weather prediction (HIRLAM; Undén et al., 2002); (ii) atmospheric chemical transport (CAMx; Ramboll Environ, 2016; Amstrup et al., 2010a; Amstrup et al., 2010b); (iii) computational fluid dynamics (M2UE; Nuterman, et al., 2010; Sabatino, et al., 2011) – were used for

airflow and pollution prediction at city/street-scale (on example of Copenhagen, Denmark). In the Methodology section, we firstly describe the applied models and their configurations; and secondly - their interplay within semi-operational setup of the downscaling system. The Results and Discussion section presents the models' evaluation against two urban observation sites (at roof and street levels) as well as recommendations about possible improvements in downscaling system to achieve better prediction quality of forecasting at finest scales.


## 2 Methodology: downscaling model system

### 2.1 Description of models

#### HIRLAM

The Numerical Weather Prediction (NWP) model used in the current downscaling system is the DMI's version of High-Resolution Limited Area Model (HIRLAM; Undén et al., 2002; Petersen et al., 2005). It has basic components such as the hydrostatic semi-Lagrangian dynamical core, digital filtering initialization (Lynch and Huang, 1992), physical parameterization of radiation processes (Savijärvi, 1990), cloud-microphysics Soft Transition Condensation scheme (Sass, 2002), turbulence parametrization (Cuxart et al., 2000), and Interaction-Soil-Biosphere-Atmosphere surface scheme (ISBA; Noilhan and Planton, 1989). The ISBA scheme also includes the building effect parametrization (BEP; Martilli et al., 2002) module to account for effects associated with urban anthropogenic heat flux and roughness. The model is forced by meteorological lateral boundary conditions and 3D-VAR assimilated ground-based and satellite observation data (Gustafsson et al., 2001; Lindskog et al., 2001). The lateral boundaries are received every 6 h interval from the European Centre for Medium-Range Weather Forecasts (ECMWF) and had horizontal grid resolution of 0.15°×0.15°.

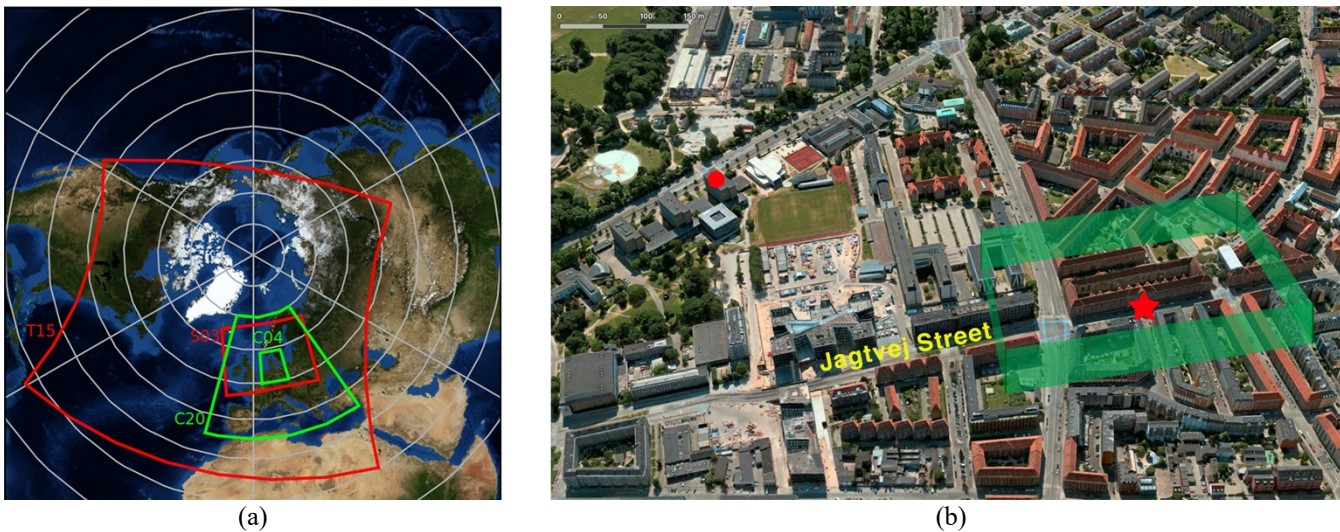

(a)                                            (b)

**Figure 1:** (a) operational and research modelling domains for European regional-scale (HIRLAM-T15+CAMx-C20) and sub-regional/urban Denmark scale (HIRLAM-S03+CAMx-C04) forecasts; (b) Nørrebro district of Copenhagen urban area with 2 air-quality monitoring stations, where red circle indicates a roof level air-quality observation at H.C. Ørsted Institute and red star indicates Jagtvej Street level observation; the green trapezoid shows M2UE modelling domain (extracted from © Apple Maps 2020).

In this study, the chosen modelling domains were used for operational numerical weather prediction at DMI. These two domains with crude (T15: 0.15°×0.15°) and fine (S03: 0.03°×0.03°) horizontal grids were applied for European regional and sub-regional/urban Denmark scale modelling (Fig. 1a), correspondingly. In vertical, 40 hybrid pressure levels (starting from

32 m above ground) were used in T15 domain and 65 such levels (starting from 12 m above ground) - in S03 domain. The upper boundary reached 10 hPa at the top of the atmosphere in both computational domains.

There are approximately 8 x 8 horizontal grids cells (each cell is about 3 x 3 km size) covering the Copenhagen metropolitan area in S03 domain. According to (Mahura et al., 2008b), the HIRLAM model with activated BEP module must have at least 4 times more grid cells to predict a significant effect of the city area on meteorology. Thus, the BEP module was disabled

during operational forecasting with S03 domain, considering the employed numerical grid as well as city size and urban characteristics of Copenhagen. The urban effects were taken into account through roughness parameter and meteorological data assimilated from urban and suburban stations.

## CAMx

The Comprehensive Air Quality Model with Extensions (CAMx; Ramboll Environ, 2016) is the finite volume ACT model for regional- and urban-scales employed for air-quality forecasting in our downscaling system. An updated version of Carbon Bond IV (CB-IV) Mechanism (Gery et al., 1989) with improved isoprene chemistry (Yarwood et al., 2005) and 2-mode aerosol microphysics scheme were used in the current setup. The Tropospheric Ultraviolet and Visible radiation model (Madronich and Focke, 1998) was applied to calculate photolysis rate coefficients. Further details about gas-phase chemistry

and aerosol microphysics together with wet (Seinfeld and Pandis, 2016) and dry (Wesely, 1989; Slinn and Slinn, 1980) deposition schemes are described in the model's users guide (Ramboll Environ, 2016).

For this study, two computational domains (Fig. 1a) were selected within the ACT model for modelling on regional- (C20: $0.2°×0.2°$) and sub-regional/urban scales (C04: $0.04°×0.04°$) forced by meteorological output (with a coupling interval of 1 hour) from HIRLAM-T15 and -S03 modelling domains, respectively. The vertical resolution of CAMx numerical grid is

defined by the one of meteorological driver and therefore, corresponds to 25 HIRLAM hybrid pressure levels covering the lowest 3 km of the troposphere. The chemical initial and boundary conditions for regional-scale modelling are represented by climatological mean values, and those for the sub-regional/urban scale modelling are obtained from the regional scale CAMx model output.

The emission rates were calculated from emission inventory provided by TNO (Kuenen et al., 2010). Although, the EMEP

emission inventory was also available, but at substantially lower resolution of $0.5°×0.5°$, and therefore, the TNO inventory was used. The inventory has longitude-latitude resolution of $0.125°×0.0625°$ and covers the entire Europe, Western Russia and Northern Africa. It includes gases ($NO_x, SO_2, CO, NMVOC, NH_3, CH_4$) and lumped primary aerosols with diameters less than 2.5 μm and 10 μm. The emission rates are associated with different anthropogenic sources attributed to various anthropogenic activities according to the Selected Nomenclature for sources of Air Pollution (SNAP).


## M2UE

For the city/street scale modelling, the Micro-scale Model for Urban Environment (M2UE) was developed and applied. It is a Computational Fluid Dynamics (CFD) code for airflow and pollution prediction in an urban environment (Nuterman et al.,

2010), which is able to simulate a complex aerodynamics in heterogeneous urban canopy with penetrable (vegetation) and impenetrable (buildings) obstacles and traffic induced turbulence. The model includes three-dimensional system of Reynolds equations, $k - \varepsilon$ (Launder and Spalding, 1974) and cubic eddy-viscosity (Craft et al., 1996) turbulence closure schemes, and the 'advection-diffusion' equations to simulate atmospheric chemistry transport. A numerical solution of governing equations is based on the fully implicit time advancing scheme and the finite volume method (Patankar, 1980; Ilin, 1995; van Leer, 1974). An ordinary uniform or non-uniform regular 3D grid with fictitious domain method (Aloyan et al., 1982; Vabishchevich, 1991) is used in the model. The M2UE model is coupled with simplified atmospheric gas-phase chemistry mechanism. This mechanism is based on Stockwell and Goliff (2002) with extra chemical reaction included: $SO_2 + OH \longrightarrow H_2SO_4 + HO_2$. Full list of chemical reactions and corresponding rate parameter for this reaction are given in Table A1, Appendix A. Such simple chemistry is dictated by the fact that the model was run in a semi-operational mode, which implies relatively short forecasting time on available computer facility. In the future, when more powerful supercomputers become available the mechanism could be extended or improved with more comprehensive one.

The computational domain (a section of Jagtvej Street and surroundings) used in the current study is explicitly defined by buildings, roads, vegetation of the Copenhagen metropolitan (Fig. 1b) area obtained from the high-resolution Danish Elevation Model dataset (Flatman et al., 2016). The horizontal and vertical extent of the computational domain is 500 x 500 x 100 m (detailed grid is shown in Fig. A2). The distance between buildings in the centre of computational domain and inlet/outlet boundaries is 5$H_{max}$, where $H_{max}$ = 20 m is the tallest building height. The distance between the tallest building and top boundary was 4$H_{max}$. Therefore, the current micro-scale domain extent/size is acceptable compromise between the best practices and guidelines (Franke et al., 2007) as well as total operational runtime (should not exceed 12 hr in order to satisfy modelling system requirements for the entire modelling chain). The horizontal and vertical numerical grids with local refinement of grid-cells ranging from 1 m at the Jagtvej Street to 5 m near the domain's boundaries was used. The Jagtvej Street emission inventory was derived from hourly traffic counts (Berkowicz et al., 2004) and emissions for different types of road-vehicles (Chao et al., 2018). The traffic volume at the Jagtvej street is similar to measured diurnal variations (Berkowicz et al., 2004; Berkowicz et al., 2006). The emissions are given in Appendix A (see Fig. A1), where only one peak of concentration was observed during morning hours and also proven by street-level observations. Note, the emissions utilized in large- and micro-scale models are different. The CAMx model uses annual emission inventory from TNO. It is pre-procced to account for monthly, weekly and diurnal emission variations. M2UE uses traffic counts measured at the Jagtvej street of Copenhagen to derive more accurate emissions (Fig. A1) and hence, account for a very local spatio-temporal features specific for this street.

Previously, the M2UE model contributed to the COST Action 732 evaluation exercise. This exercise was devoted to micro-scale models quality evaluation and assurance. Moreover, the dispersion patterns of chemical species in typical urban canyons were investigated in the previous studies (Franke et al., 2007; Schatzmann et al., 2010; Nuterman et al., 2010; Nuterman et al., 2011; Sabatino et al., 2011) and would not add value to this research. Since the model's dispersion capabilities were thoroughly tested, it seemed interesting and valuable to focus on operational aspects of the problem, and

hence, to apply the CFD M2UE model using real time weather and atmospheric composition fields. Therefore, the purpose of the current study is rather the concept proof (on possibility to run CFD type model operationally coupled with gas-phase chemistry) than rigorous full-scale evaluation of the proposed downscaling chain.

## 2.2 Description of downscaling model system

In the developed downscaling model system (Fig. 2), both the European regional (hereafter referred to as the EU) and the sub-regional/urban Denmark scale (hereafter referred to as the DK) forecasts were performed by the off-line coupled HIRLAM and CAMx models. After up-to-date meteorological data assimilation and boundary conditions from ECMWF became available, the HIRLAM-T15 produced 48 h meteorological forecast for the EU domain. This NWP model was always fully spun-up, because such forecast was performed in a continuous manner 4 times per day (at 00, 06, 12 and 18 UTC). Afterwards, the meteorological output was post-processed, interpolated and adapted to drive the CAMx-C20 model with a spin-up time of 48 hour and the forecast length of 24 hour. The spin-up integration was the model's cold-start run prior to the current forecasted day. The entire cycle of offline coupled HIRLAM-T15+CAMx-C20 modelling chain took approximately 4.5 hour of CPU time.

As soon as the EU scale forecast is completed, the DK scale forecast of meteorology and atmospheric composition is initiated. In general, the entire procedure of model runs and interplay is very similar to the EU modelling chain. However, the differences were in boundary conditions, forcing and resolution of numerical grids. The HIRLAM-S03 meteorological forecast was forced by outer model lateral boundaries, while the CAMx-C04 atmospheric composition forecast was forced by meteorology from HIRLAM-S03 plus chemical boundary and initial conditions from CAMx-C20. For each integration cycle, new boundary fields are read at 1-h interval. So, the models retrieve updated meteorological and chemical fields from the previous low-resolution outer domains. The forecast length of HIRLAM-S03 and CAMx-C04 runs was 36 h and 24 h, respectively. The forecast cycle used 4 h of CPU time since the DK forecast initiation.

After the urban-scale forecast completed, the street-scale (hereafter referred to as the street) forecast is launched for a section of Jagtvej Street of Copenhagen and surroundings. For the Jagtvej street domain inlet boundaries, the Dirichlet conditions with constant profiles of airflow characteristics (from HIRLAM-S03), averaged urban roughness and atmospheric composition fields (from CAMx-C04) were chosen to drive the M2UE model. In particular, the output from S03 and C04 was linearly interpolated from four surrounding grid points into a grid point corresponding to the M2UE domain. The interpolated fields were used to derive the mean inlet conditions under the assumption of equilibrium boundary layer with power-law velocity profile and local equilibria assumption for turbulent parameters (Richards and Hoxey, 1993). As for the chemical fields, the interpolated values were considered as the urban background concentrations. The outlet boundary condition is of the Neumann type with zero normal derivatives of all meteorological and chemical concentration variables corresponding to a fully developed and well-mixed flow. These above-mentioned boundary conditions follow the COST Action 732 best practises and guidelines (Franke et al., 2007) for micro-scale meteorological and atmospheric composition models and ensure consistency with each other. The forecast for 24 h was performed after HIRLAM-S03+CAMx-C04

completed runs and it took ~1.5 h of CPU time. Since the M2UE model is very computationally expensive, the 24 h forecast was performed with 24 instant integration cycles forced by hourly HIRLAM-CAMx models' output. In this case, the spin-up time was just several minutes; so, such as for individual M2UE forecast. It is a reasonable approach of doing such the forecast, because the boundary conditions and emission rates for each integration cycle were constant and only the rapid-

190 photochemical reactions of $O_3$ and $NO_x$ were of principal interest in the study. Due to the fact that the nested M2UE domain and roof-level observation site are spatially located very close to each other and within the same grid-cell of DK domain, the time-series of CFD boundary conditions are identical for $O_3$ and $NO_x$. Therefore, the air-flow regime was assumed to be steady during an instant integration cycle. The ratio of horizontal resolution of HIRLAM-S03+CAMx-C04 and M2UE-Jagtvej domains exceeds a permissible limit of 5 (Sokhi et al., 2018). However, the runtime of the entire modelling chain

equals to 10 CPU hours, and adding an intermediate domain (or chain of domains) between these 3 model domains will obviously lead to increased runtime over 12 hours. Such large span of forecasting time is considered to be too large, taking into account the time constrains for operational tasks and cases when the models' rerun is required (not to mention pre-processing and post-processing the large amount of models' input and output). Thus, such downscaling is a trade-off between acceptable runtime and permissible resolution.

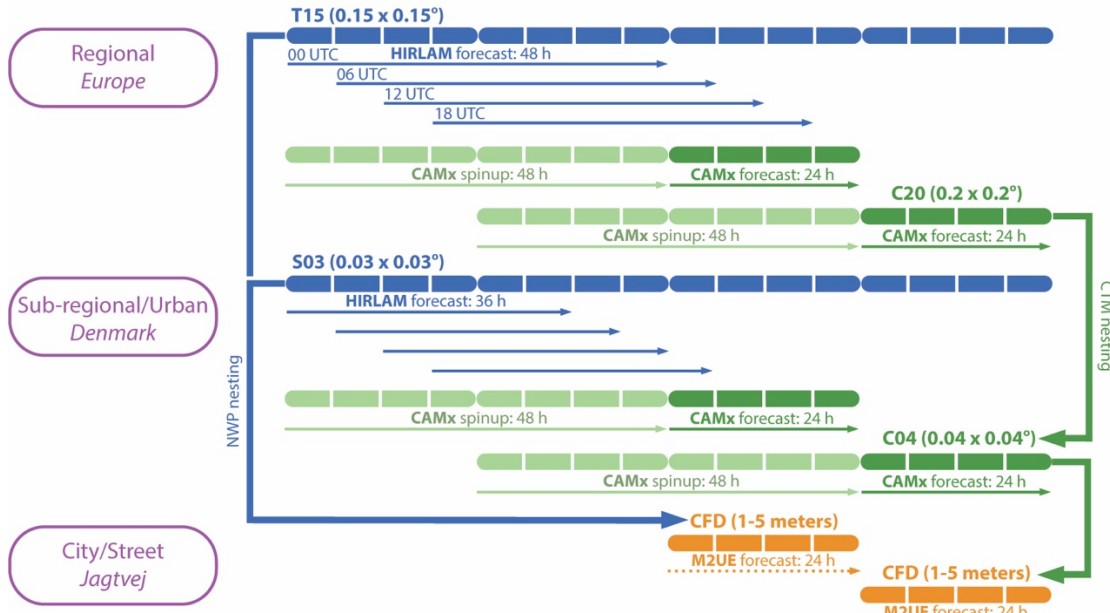

**Figure 2:** Downscaling modelling system: from the European regional-scale forecast (HIRLAM-T15+CAMx-C20) to the sub-regional/urban Denmark (HIRLAM-T03+CAMx-C04) and city/street scale for Copenhagen (CFD M2UE); the blue, green and orange horizontal bars indicate NWP HIRLAM, CTM CAMx and CFD M2UE forecasts, respectively; horizontal and vertical arrows indicate the

205 forecasts' cycles and downscaling (dataflow), correspondingly.

## 3 Results and Discussion

The example of the downscaling modelling chain realised in this study in a semi-operational mode with coupled HIRLAM+CAMx+M2UE models is described and evaluated with focus on Denmark and Copenhagen for specific meteorological situation.

During 4-5 September 2011 the meteorological situation was the following. The cyclonic system originated over Iceland moved southward towards the British Islands, and then eastward towards the Jutland Peninsula of Denmark with gradually changing atmospheric pressure from 990 to 1010 hPa. The cold front associated with cyclone passed over UK and Denmark on these days. On 4th September morning, the cold front approached the northern part of the Jutland Peninsula, and then propagated towards the Zealand Island, and brought up to 10 mm of accumulated (6 hour) precipitation. Over Denmark, during night-morning hours on 5th September, the precipitation increased on average up to 15-20 mm (with several local maxima up to 40 mm). During these days, on a diurnal cycle the air temperature's range was 18-24 °C and 12-18 °C (on 4th and 5th September, respectively) with further gradual cooling. The vertical sounding diagrams (Schleswig, Germany, 54.53° N, 9.55° E – the closest station to Denmark) showed the presence of the temperature inversion layer extended up to 1.5 km (at noon on 4th Sep) and reduced to 500 m altitude (at night on 5th Sep), and then a layer of smaller depth remained at top of boundary layer at noon.

A near surface spatial distribution of $O_3$ and $NO_x$ concentrations (Fig. 3ab) within the EU domain is shown for 5th Sep 2011, at 10 UTC. As seen, the elevated $NO_x$ concentration of up to 50 µg m$^{-3}$ was observed over urban, industrial and port areas of Denmark and neighbouring countries. In addition, the North Sea oil extraction and international shipping were other sources of elevated $NO_x$ levels over the Danish territorial waters. The near surface $O_3$ concentration varied within the range of 40-150 µg m$^{-3}$, with local minima over and downwind the areas of anthropogenic $NO_x$ sources (Fig. 3a) where ozone was depressed due to the $NO_x$ titration.

As for the nested DK domain (Fig. 3cd), it is seen from the near-surface wind and concentrations patterns that the mesoscale circulation as well as the local anthropogenic emission sources were resolved in greater details due to the finer S03-C04 horizontal resolution. Therefore, one can observe additional local concentration maxima of $NO_x$ with higher levels up to 250 µg m$^{-3}$ over Denmark. These were mostly localized near the same type of anthropogenic sources as the mentioned above in the EU domain. As the result, there were more $O_3$ depression spots including those over the northern Jutland Peninsula's highways over the territories of Denmark and Germany. However, the background values of the near surface ozone did not change significantly at large distances from the anthropogenic $NO_x$ sources.

In Figure 4, the time-series of $O_3$ and $NO_x$ observations provided by the Danish air-quality monitoring programme (Ellerman et al., 2012) and modelled concentrations by the downscaling system are presented. Those observations were performed by automatic stations (Fig. 1b) at the roof level of H.C. Ørsted Institute and at the Jagtvej Street (only $NO_x$, particulate matter and benzene were measured on the street; ozone and carbon monoxide were not available for that time at this location). As seen in Fig. 4a, the T15-C20 and S03-C04 results showed an ability to reproduce well the $O_3$ diurnal cycle ($EU\_R_p = 0.81$

and $DK\_R_p = 0.8$, with p $\ll$ 5 %) for both domains at regional- and sub-regional/urban scales. However, the forecasted $O_3$

at the EU scale was generally overestimated in comparison to the modelling results at the DK scale ($EU\_RMSE = 31$ vs. $DK\_RMSE = 19$). Although, the $NO_x$ levels were generally overestimated in the DK scale forecast (Fig. 4b) when compared to the EU scale forecast and observations ($EU\_RMSE = 23$ vs. $DK\_RMSE = 47$), the diurnal cycle at the DK scale is better captured according to obtained Pearson correlation values ($EU\_R_p = 0.56$ vs. $DK\_R_p = 0.69$). And it is particularly notable when the $NO_x$ peak was observed during several morning hours (on 5th Sep 2011, 07 UTC). This caused by the complex

unfavourable meteorological conditions over the Copenhagen metropolitan area and high emission rates mainly from intense morning traffic. The timing and levels of $NO_x$ peak were well captured by the DK scale forecast in contrast to the EU one, which poorly predicted it with slightly elevated $NO_x$ values. Note, the $O_3$ depression at the same time, which was not captured by the EU air-quality forecast. Since the primary sources of atmospheric nitrogen are anthropogenic emissions, its overestimated concentrations are rather attributed to pre-processing and treatment of emission rates. It can be related to

interpolation of original gridded emissions into the meshes of ACT model, which is apparently insufficient, and therefore, some additional rescaling adjustments in terms of emission rates and their time dynamics are required.

The diurnal cycle of $NO_x$ concentrations observed and modelled for the Jagtvej Street is shown in Fig. 4c. Following analysis and calculated correlation value ($R_p = 0.66$), the street-scale M2UE model forecasted well the observed $NO_x$ diurnal cycle including the onset of elevated air pollution episode on 5th Sep (see forecast length 30-35 h). It corresponded to

the local time of 8-10 h of a typical working day (Monday) when the most intense road traffic is observed. However, the model's quantitative estimate of concentration is substantially biased (with $RMSE = 103$) particularly during 5th Sep 2011. In spite of the captured elevated $NO_x$ levels, the model predicted two peaks at 31 and 34 h of forecast length and depression at 32 h, while the time-series of observations showed only one peak at 32 h, which was approximately 200 µg m$^{-3}$ higher than the modelled peaks.

There are several reasons why the M2UE model was able to make the qualitatively good prediction in general but had difficulties at capturing the elevated $NO_x$ concentrations. First of all, it is the coupling interval span of the offline coupled models (HIRLAM+CAMx), which plays an important role in constraining the influence of mesoscale disturbances on evolution of atmospheric composition (Korsholm et al., 2011). It might give an offset in timing and position of the predicted concentrations. As seen in Fig. 4ab, the effect of that at the onset of the high air pollution episode (where its slight offset of

approximately half an hour) was observed, and then inherited by the nested micro-scale model. The second reason is that the M2UE forcing was provided on an hourly basis by the HIRLAM-S03+CAMx-C04 modelling chain. Such forcing is rather crude in terms of time and horizontal resolution for the CFD model capable to predict very fine spatio-temporal turbulent flow scales occurring within meters and seconds. The third reason is an uncertainty in local road traffic emissions used by the M2UE model, although some traffic count data and information about emissions from different types of vehicles were

available (Chao et al., 2018). Although those above-mentioned shortcomings, the presented downscaling modelling system was able to produce good quality atmospheric composition forecasts across different spatial and temporal scales.

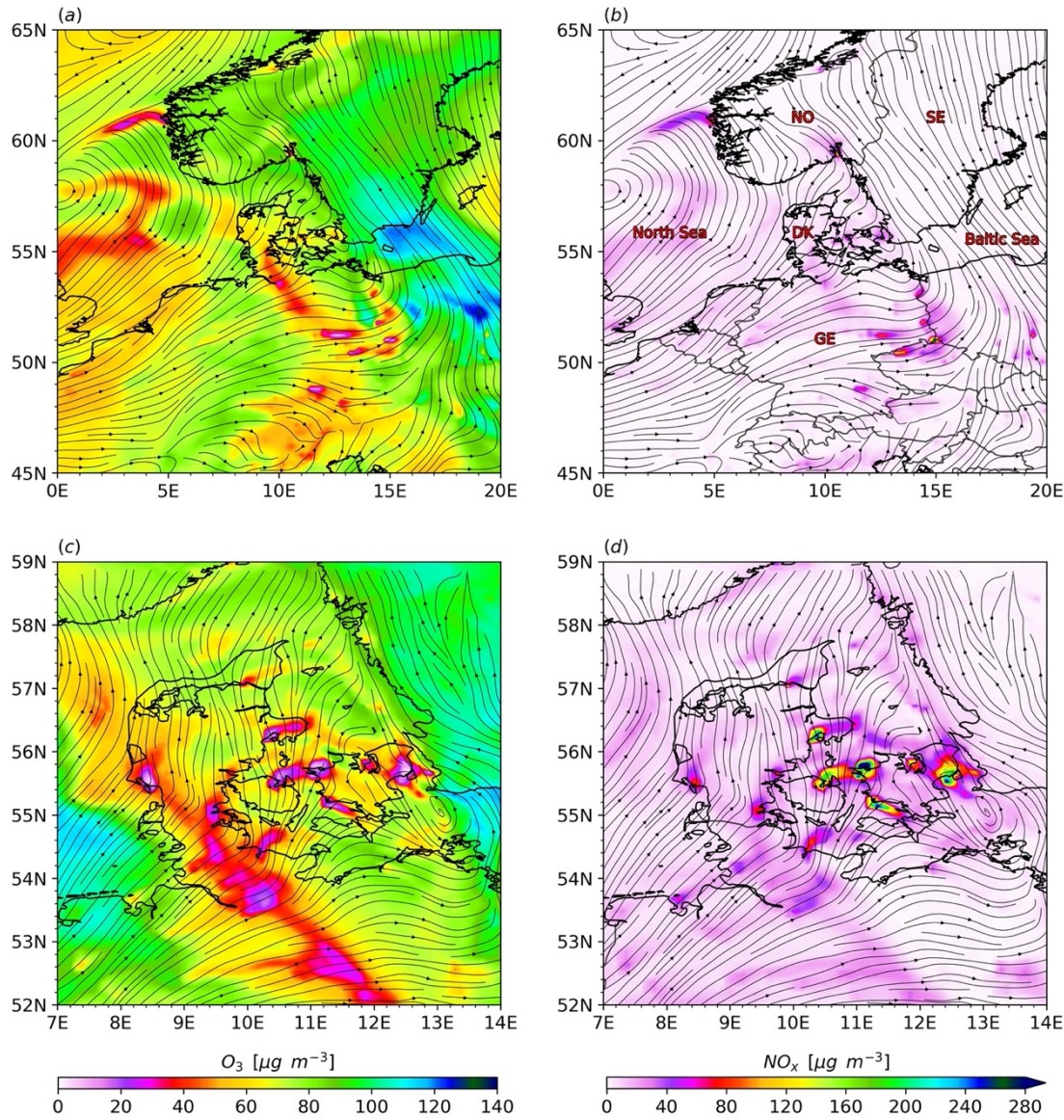

**Figure 3:** Simulation output on 5 Sep 2011, 10 UTC: HIRLAM-T15+CAMx-C20 results for concentrations in [μg m⁻³] of (a) $O_3$ and (b) $NO_x$ within the European regional-scale domain and HIRLAM-S03+CAMx-C04 results for concentrations of (c) $O_3$ and (d) $NO_x$ within the Denmark sub-regional/urban scale domain.

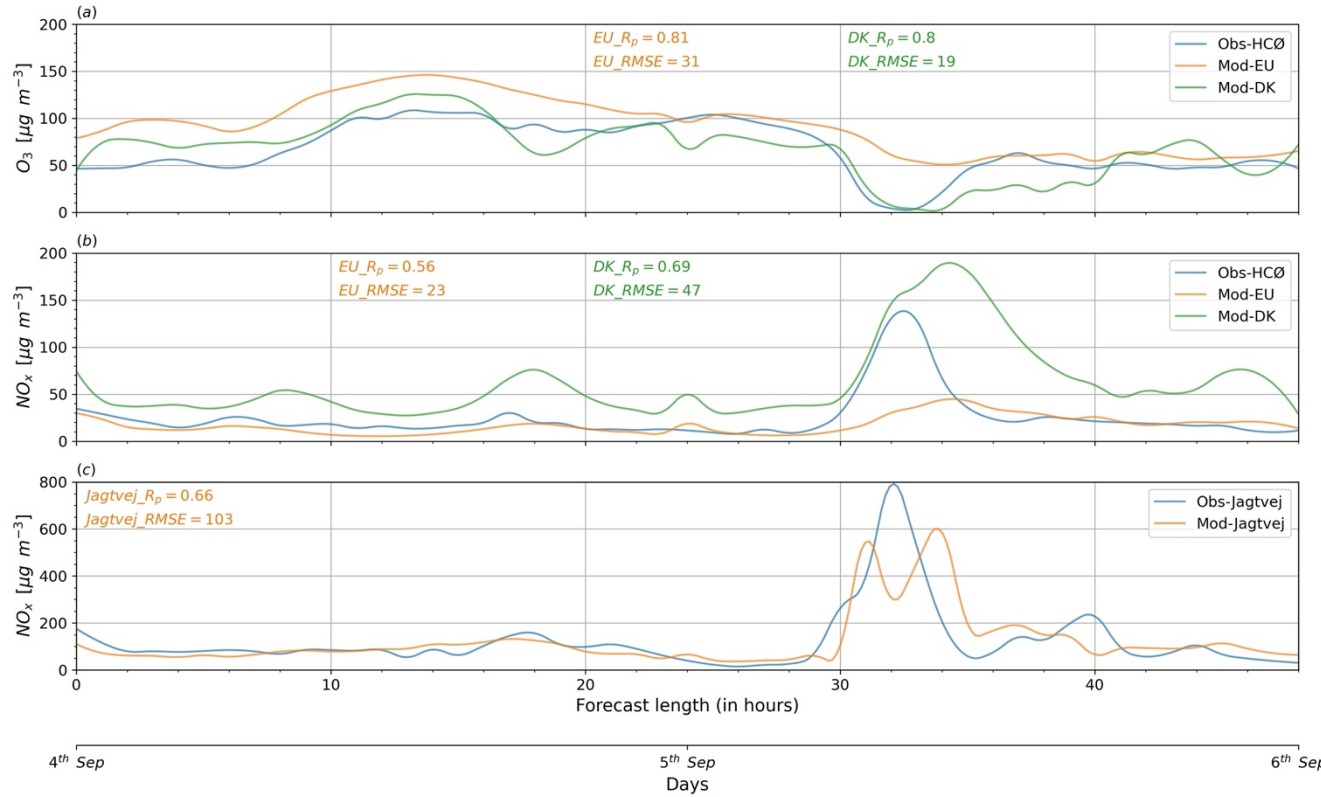

**Figure 4:** Time-series of observations at the building's roof level vs. HIRLAM+CAMx+M2UE forecasts for (a) $O_3$ and (b, c) $NO_x$ for the
European, Denmark and Copenhagen modelling domains during Sunday-Monday 4-5 Sep 2011; (a, b) HCØ roof level observations of $O_3$
and $NO_x$ vs. modelling results for EU and DK domains; (c) Jagtvej Street level observations of $NO_x$ vs. M2UE /Pearson correlation $R_p$
and root mean square error RMSE of modelled vs. observed concentrations for 3 domains/.

## 4 Conclusions

In this study, the downscaling modelling system for real time prediction of weather and atmospheric composition up to the
street scale has been described and tested. It consists of several interfaced models. The forecasts were performed at the
regional and sub-regional/urban-scales by the offline coupled operational numerical weather prediction HIRLAM and
atmospheric chemical transport CAMx models and at the city/street scale by the online coupled computational fluid
dynamics M2UE model. The downscaling system was tested for the Europe (EU), Denmark (DK) and Copenhagen's street
domains in a semi-operational forecasting mode and evaluated against existing observations for a high pollution episode of
4-5 September 2011.

The results of modelling showed elevated $NO_x$ and lowered $O_3$ concentrations over major urban, industrial and transport
land/water routes in both EU and DK scale forecasts. The comparison of these forecasts showed higher levels of $NO_x$ and
slightly lower $O_3$ in the DK domain because of higher $NO_x$ emission rates and titration. The $O_3$ diurnal cycle predictions in
both the EU and DK domains are equally good, although $O_3$ values were closer to observations for DK. At the same time,

the DK scale forecast of $NO_x$ levels was more biased (with better prediction score of the diurnal cycle) than EU scale forecast. Further downscaling to the street level indicated that the $NO_x$ pollution (mainly from road traffic) was 2-fold higher on weekend and more than 5 times higher during working day with high pollution episode. Despite of high uncertainty in road traffic emissions, the M2UE model captured well the $NO_x$ diurnal cycle and onset of elevated pollution episode. However, in order to come to the final conclusion about the performance of individual system's components and the entire

downscaling chain, one needs to perform an evaluation for extended time period and additional measurement sites around Denmark.

  A potential improvement of such downscaling system can come from employing online coupled (instead of offline coupled) mesoscale models. Such approach could help to minimize problems associated with offline coupling interval and therefore, to give a better fit of modelling results to observations, and to better identify a timing and location of elevated pollution

episodes. As for the local-scale predictions, an intermediate additional nested domain between sub-regional/urban and micro-scale runs for the Copenhagen metropolitan area could provide improvement for local scale airflow and pollution forecasting scores. However, these will be conditioned on availability of a better-quality high-resolution emission inventory. The demonstrated downscaling system with the suggested above improvements could be used in future online integrated meteorology and air quality research and operational forecasting. Such system can be also applied for advanced planning

safety measures, post-accidental analysis, emergency preparedness and health/environment impact assessments.

  Keeping in mind the main uncertainties (e.g., street-level emissions, increasing number of urban observations, etc.), there are several possibilities to improve demonstrated urban / street level forecasts using different types of observations (e.g., from remote sensing, crowed sourcing and citizen science) in urban areas. These observations and recently developing methods such as the bias correction techniques, machine learning, artificial intelligence and neural networks can also improve air-

quality forecasting (see overview e.g., in Bai et al., 2018).

**Appendix A**

Table A1: List of chemical reactions included as gas-phase chemistry mechanism in M2UE model

| | |
|---|---|
| $NO_2 + hv \longrightarrow O(^3P) + NO$ | $RO_2 + NO \longrightarrow NO_2 + HO_2 + HCHO$ |
| $O_3 + hv \longrightarrow O(^1D) + O_2$ | $CO + HO \longrightarrow HO_2 + CO_2$ |
| $HCHO + hv \longrightarrow 2HO_2 + CO$ | $HO + HC \longrightarrow RO_2 + H_2O$ |
| $HCHO + hv \longrightarrow H_2 + CO$ | $HO + HCHO \longrightarrow HO_2 + CO + H_2O$ |
| $O(^3P) + O_2 \longrightarrow O_3$ | $HO + NO_2 \longrightarrow HNO_3$ |
| $O(^1D) + N_2 \longrightarrow O(^3P) + N_2$ | $2HO_2 \longrightarrow H_2O_2 + O_2$ |
| $O(^1D) + O_2 \longrightarrow O(^3P) + O_2$ | $2HO_2 + H_2O \longrightarrow H_2O_2 + H_2O + O_2$ |
| $O(^1D) + H_2O \longrightarrow 2OH$ | $HO_2 + RO_2 \longrightarrow ROOH + O_2$ |
| $HO_2 + NO \longrightarrow NO_2 + HO$ | $2RO_2 \longrightarrow 2HCHO + HO_2$ |

| $O_3 + NO \rightarrow NO_2 + O_2$ | $SO_2 + OH \rightarrow H_2SO_4 + HO_2$ |
|---|---|

Chemical reactions and rate parameters are the same as in Stockwell and Goliff (2002), except chemical reaction ($SO_2 +$

$OH \rightarrow H_2SO_4 + HO_2$) with the rate parameter from (Sander et al., 2006):

$$k_0 = 3 \times 10^{-31} \times C_{air} \times (300/T)^{3.3}, k_\infty = 1.5 \times 10^{-12}$$

$$k = \frac{k_0}{1 + k_0/k_\infty} \times 0.6^{(1+[\log_{10}(k_0/k_\infty)]^2)^{-1}}$$

where $T$ is temperature [K] and $C_{air}$ is air concentration [molecules/cm³].

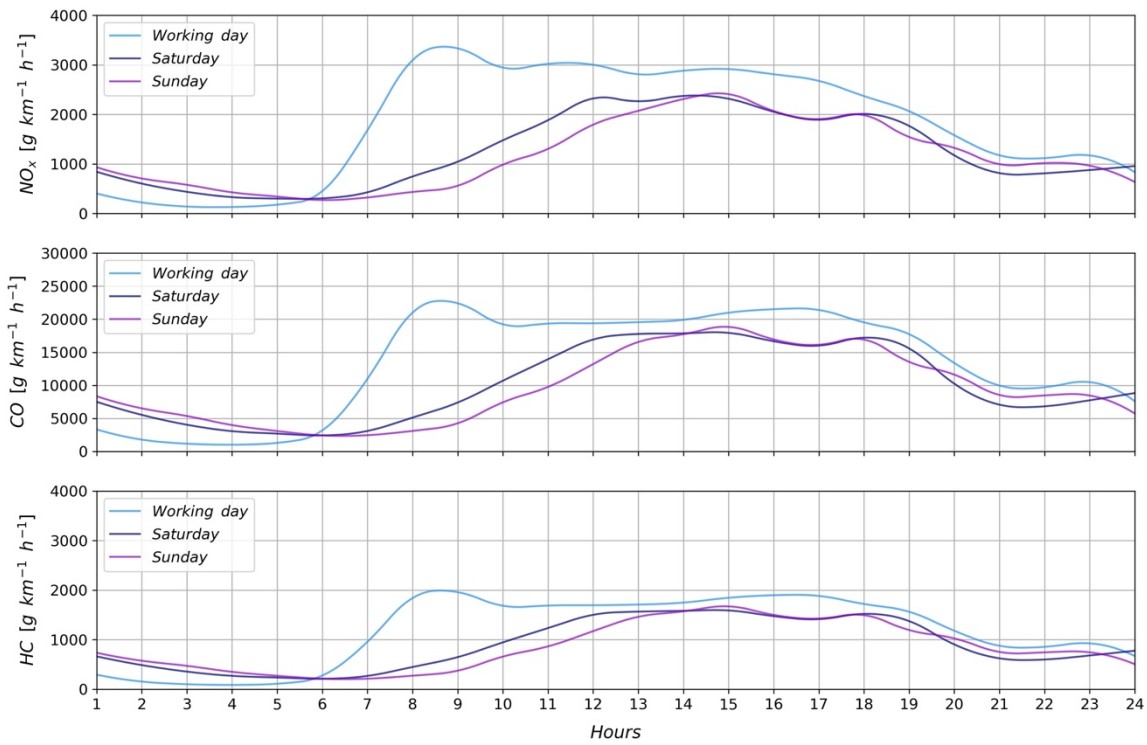


Figure A1: Time-series of diurnal cycle emissions on typical Saturday, Sunday and Working day of a week at the Jagtvej street, Copenhagen, Denmark.

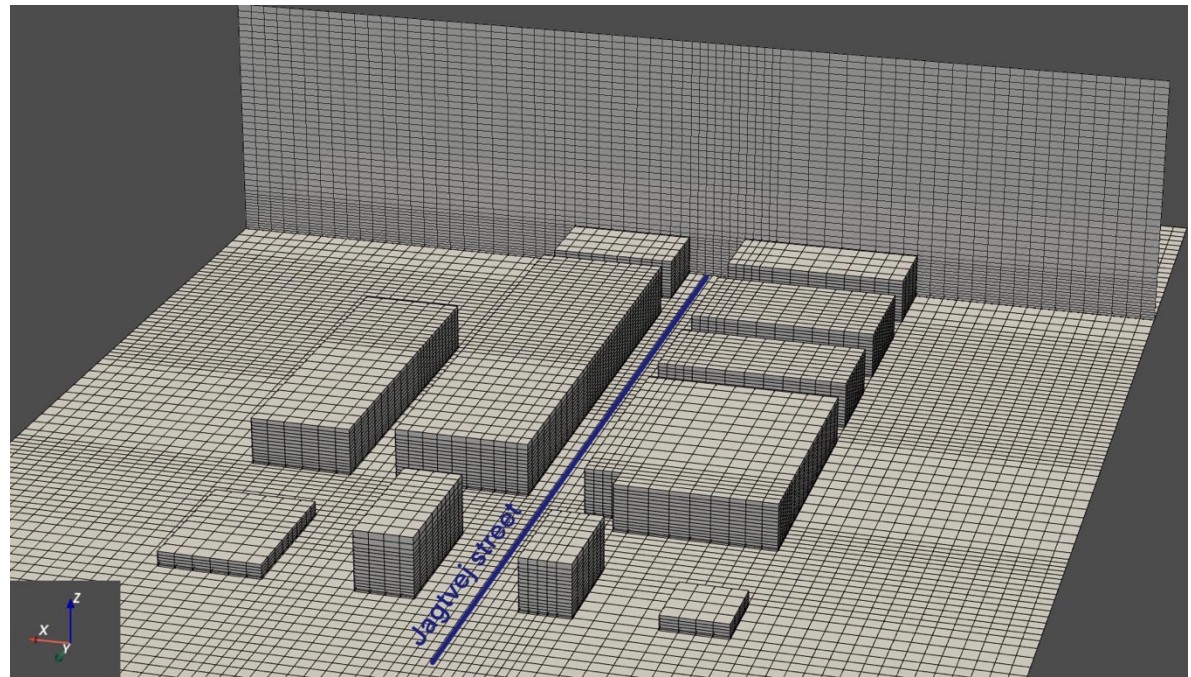

Figure A2: M2UE numerical grid of the Jagtvej street, Copenhagen, Denmark.

## Code availability

The HIRLAM model source code is available under the HIRLAM consortium agreement (http://hirlam.org). The CAMx model source code is an open-source software (can be downloaded from http://www.camx.com). The M2UE model source code is available from the authors upon request.

**Authors contribution**

AB, AM designed the research; BA, RN created and setup integrated downscaling of the models; AZ developed local scale gas-phase chemistry mechanism; RN performed models' evaluation; RN, AM and AB wrote the paper.

## Competing interests

The authors declare that they have no conflict of interest.

## Acknowledgments

The research leading to these results received funding from the European Union's Seventh Framework Programme FP/2007-2011 under grant agreement n°218793 (MACC – Monitoring of Atmospheric Composition and Climate; 2009-2011). The authors are thankful to the PEEX (Pan-Eurasian Experiment; https://www.atm.helsinki.fi/peex) programme for long-term collaboration within the PEEX's Modelling Platform and Impact on Society tasks. The authors are thankful to Drs. Bent H. Sass and Jens H. Sorensen (Danish Meteorological Institute, DMI; Copenhagen, Denmark) for administrative support. Especial thanks to DMI's IT and eScience departments for computing advise and support on HPC CRAY-XT5, and in particular to Dr. Jacob Weismann (DMI) for the M2UE model optimization.

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
