# Peer review of "Downscaling system for modelling of atmospheric composition on regional, urban and street scales"

_Atmospheric Chemistry and Physics, 2020_

## Author Comment (AC1)

**Author responses to reviewers' comments of ACP-2020-1308 - Downscaling system for modelling of atmospheric composition on regional, urban and street scales**

*Roman Nuterman, Alexander Mahura, Alexander Baklanov, Bjarne Amstrup, and Ashraf Zakey*

We are grateful to the reviewers for critical and valuable comments. Below we provide the response to reviewer's comments and suggestions (with table and figures included in the Appendix A).

**Reviewer #1**

**Question - Line 128**. In the M2UE simulation, is the 500 m horizontal extent sufficient to represent the atmospheric environment of the street? I wonder if the airflow around the monitoring site is still substantially influenced by the boundary conditions. The extent of the airflow around the street that is largely affected by boundaries needs to be further investigated.

**Answer**: The M2UE computational domain was constructed following the COST Action 732 best practices and guidelines (see Franke et al., 2007) to ensure minimal influence of inflow/outflow boundaries on the airflow and pollution dispersion. Therefore, for the current micro-scale simulations with a steady-state RANS approach, the distance between buildings and inlet/outlet boundaries was equal to $5H_{max}$, where $H_{max} = 20$ m is the tallest building height. The distance between the tallest building and top boundary was $4H_{max}$. The computational domain could have been larger in extent, in particular, towards the outflow boundary. However, that would not satisfy the operational modelling system requirements because the total runtime interval of the entire modelling chain would exceed 12 hr. Therefore, the current micro-scale domain extent/size is an acceptable compromise between the best practices and guidelines as well as total runtime as requirements.

**Hence, manuscript text was modified by adding the following sentences**: "The horizontal and vertical extent of the computational domain is 500 x 500 x 100 m (detailed grid is shown in Figure A2). The distance between buildings in the centre of computational domain and inlet/outlet boundaries is $5H_{max}$, where $H_{max} = 20$ m is the tallest building height. The distance between the tallest building and top boundary was $4H_{max}$. Therefore, the current micro-scale domain extent/size is acceptable compromise between the best practices and guidelines (Franke et al., 2007) as well as total operational runtime (should not exceed 12 hr in order to satisfy modelling system requirements for the entire modelling chain)."

**Question - Line 122**. What chemical components and their reactions are included in the M2UE? Are the emission data used in the M2UE simulation equal to the emissions in the corresponding grids in the CMAx simulation? If not, how the difference between the emission data affects the simulation results?

**Answer**: The gas-phase chemistry mechanism (see Table A1, Appendix A) is mostly identical to the mechanism described by Stockwell and Goliff (2002) in Table 2. The only difference is the following additional chemical reaction: $SO_2 + OH \longrightarrow H_2SO_4 + HO_2$ with the rate parameter from Sander et al. (2006).

The emissions in large- and micro-scale models are different. The CAMx model uses annual emission inventory from TNO, which is pre-procced before simulations to account for monthly, weekly and diurnal variations. The M2UE model uses traffic counts measured at the Jagtvej street of Copenhagen in order to derive emissions, and therefore, are considered to be more accurate. Moreover, it would be rather crude to use TNO emissions for the micro-scale modelling as these do not account for a very local spatio-temporal features specific for this street.

**Hence, manuscript text was modified by adding the following sentences**: "The M2UE model is coupled with simplified atmospheric gas-phase chemistry mechanism. This mechanism is based on Stockwell and Goliff (2002) with extra chemical reaction included: $SO_2 + OH \longrightarrow H_2SO_4 + HO_2$. Full list of chemical reactions and corresponding rate parameter for this reaction are given in Table A1, Appendix A."

**Line 131:** "Note, the emissions utilized in large- and micro-scale models are different. The CAMx model uses annual emission inventory from TNO. It is pre-procced to account for monthly, weekly and diurnal emission variations. M2UE uses traffic counts measured at the Jagtvej street of Copenhagen to derive more accurate emissions (Figure A1) and hence, account for a very local spatio-temporal features specific for this street."

**Question - Line 150**. Are the inlet or outlet properties of each boundary fixed in the 48 h simulations? How many grids in HIRLAM-S03 and CMAx-04 are chosen to drive the M2UE model? Are the inlet/outlet properties derived from these coarse grids consistent with each other?

**Answer**: During each model integration cycle new boundary fields are read at 1-h interval, so the models retrieve updated meteorological and chemical fields from the previous low-resolution outer domains HIRLAM-T15 and CAMx-C20.

In order to drive the M2UE model, the output from the HIRLAM-S03 and CAMx-C04 models is linearly interpolated from four surrounding grids points into a grid point corresponding to the centre of the M2UE domain. The interpolated fields are used to derive the mean inlet conditions under the assumption of equilibrium boundary layer with power-law velocity profile and local equilibria assumption for turbulent parameters (Richards and Hoxey, 1993). As for the chemical fields the interpolated values are considered as the urban background concentrations. The outlet boundary condition are zero normal derivatives of all meteorological and chemical concentration variables corresponding to a fully developed and well-mixed flow. The above-mentioned boundary conditions follow the COST Action 732 best practises and guidelines (Franke et al., 2007) for micro-scale meteorological and atmospheric composition models and ensure consistency with each other.

**Hence, manuscript text was modified by adding the following sentences**:

**Line 146**, before "The forecast length of…": "For each integration cycle, new boundary fields are read at 1-h interval. So, the models retrieve updated meteorological and chemical fields from the previous low-resolution outer domains."

**Line 150:** "For the Jagtvej street domain inlet boundaries, the Dirichlet conditions with constant profiles of airflow characteristics (from HIRLAM-S03), averaged urban roughness and atmospheric composition fields (from CAMx-C04) were chosen to drive the M2UE model. In particular, the output from S03 and C04 was linearly interpolated from four surrounding grid points into a grid point corresponding to the M2UE domain. The interpolated fields were used to derive the mean inlet conditions under the assumption of equilibrium boundary layer with power-law velocity profile and local equilibria assumption for turbulent parameters (Richards and Hoxey, 1993). As for the chemical fields, the interpolated values were considered as the urban background concentrations. The outlet boundary condition is of the Neumann type with zero normal derivatives of all meteorological and chemical concentration variables corresponding to a fully developed and well-mixed flow. These above-mentioned boundary conditions follow the COST Action 732 best practises and guidelines (Franke et al., 2007) for micro-scale meteorological and atmospheric composition models and ensure consistency with each other."

**Question - Line 175**. Why do the authors select 2011 as the study period?

**Answer**: The year of 2011 was chosen because the research leading to the current results was a part of the FP7 EU project "Monitoring of Atmospheric Composition and Climate" (MACC; 2009-2011). This fact has been mentioned in the Acknowledgments section of the paper.

**Question - Figure 4.** Why only the NOx concentration is validated in the M2UE simulation? More chemical species are suggested to be evaluated against obervations.

For the M2UE modeling, simulation output is analyzed at only one site. Since the results of a single grid may not be enough to support the conclusion, it is recommended to add more analysis on the horizontal and vertical planes.

**Answer**: When the current results were obtained only the NOx was measured (apart from particulate matter) at the Jagtvej street measurement site. There are no measurements of ozone and carbon monoxide available for that time at this location. Previously, the M2UE model contributed to the COST Action 732 evaluation exercise, which was devoted to micro-scale models quality evaluation and assurance. Moreover, the dispersion patterns of chemical species in typical urban canyons were investigated in the previous studies (Franke et al., 2007; Schatzmann et al., 2010; Nuterman et al., 2010; Nuterman et al., 2011; Sabatino et al., 2011) and would not add value to this research. Since the model's dispersion capabilities were thoroughly tested, it seemed interesting and valuable to focus on operational aspects of the problem, and hence, to apply the CFD M2UE model using real time weather and atmospheric composition fields. Therefore, the purpose of the current paper is rather the concept proof (on possibility to run CFD type model operationally coupled with gas-phase chemistry) than rigorous full-scale evaluation of the proposed downscaling chain. To the best of the authors' knowledge, such attempts have not been practically done.

**Hence, manuscript text was modified by adding the following sentences**:

**Line 202:**"and benzene were measured on the street; ozone and carbon monoxide were not available for that time at this location)."

**After line 131:** "Previously, the M2UE model contributed to the COST Action 732 evaluation exercise. This exercise was devoted to micro-scale models quality evaluation and assurance. Moreover, the dispersion patterns of chemical species in typical urban canyons were investigated in the previous studies (Franke et al., 2007; Schatzmann et al., 2010; Nuterman et al., 2010; Nuterman et al., 2011; Sabatino et al., 2011) and would not add value to this research. Since the model's dispersion capabilities were thoroughly tested, it seemed interesting and valuable to focus on operational aspects of the problem, and hence, to apply the CFD M2UE model using real time weather and atmospheric composition fields. Therefore, the purpose of the current study is rather the concept proof (on possibility to run CFD type model operationally coupled with gas-phase chemistry) than rigorous full-scale evaluation of the proposed downscaling chain."

**Question - Line 222 and Figure 4.** Time-series data/figures of traffic volumes, emissions and boundary conditions are suggested to be provided. Do these data also show two peaks at 31 and 34 h? If so, the interesting difference between the observed concentrations and emissions may need to be discussed.

**Answer**: Time-series of M2UE boundary conditions are identical to $O_3$ and $NO_x$ values from DK domain in Figure 4, because both nested M2UE domain and roof-level observation site are spatially located very close to each other and within one grid-cell of DK domain. As for the traffic volume at the Jagtvej street, it is identical to diurnal variations measured during the EU TMR TRAPOS project (Berkowicz et al., 2004; Berkowicz et al., 2006).

The emissions are shown in Figure A1, according to which only one concentration peak was observed during morning hours and also proven by street-level observations.

**Hence, manuscript text was modified by adding the following sentences**:

**Line 196:** "Due to the fact that the nested M2UE domain and roof-level observation site are spatially located very close to each other and within the same grid-cell of DK domain, the time-series of CFD boundary conditions are identical for $O_3$ and $NO_x$."

**Line 140:** The traffic volume at the Jagtvej street is similar to measured diurnal variations (Berkowicz et al., 2004; Berkowicz et al., 2006). The emissions are given in Appendix A (see Figure A1), where only one peak of concentration was observed during morning hours and also proven by street-level observations."

**Question - Line 246.** Typo "/Pearson".
**Answer:** The typo has been corrected.

**Question - Line 247.** Typo "domians/".
**Answer:** The typo has been corrected.

**Question - Typos:** "μg" instead of "ug".
**Answer:** The typos have been corrected.

**Additional references**

Berkowicz, R., Winther, M., Ketzel, M.: Traffic pollution modelling and emission data, Environmental Modelling & Software, 21, 454-460, https://doi.org/10.1016/ j.envsoft.2004.06.013, 2006.

Franke J., Hellsten, A., Schlünzen, H., Carissimo, B. (Eds.): Best Practice Guideline for the CFD Simulation of Flows in the Urban Environment: COST Action 732 Quality Assurance and Improvement of Microscale Meteorological Models, COST Office, ISBN: 3-00-018312-4, 2007.

Nuterman, R., Starchenko, A.V., Baklanov, A.: Numerical model of urban aerodynamics and pollution dispersion. Int. J. of Environment and Pollution, 44(1). https://doi.org/10.1504/IJEP.2011.038440, 2011.

Richards, P.J. and Hoxey, R.P.: Appropriate boundary conditions for computational wind engineering models using the k-ε turbulence model, Journal of Wind Engineering and Industrial Aerodynamics, 46 & 47, 145-153, https://doi.org/10.1016/0167-6105(93)90124-7, 1993.

Sander, S. P., Friedl, R. R., Golden, D. M., Kurylo, M. J., Moortgat, G. K., Keller-Rudek, H., Wine, P. H., Ravishankara, A. R., Kolb, C. E., Molina, M. J., Finlayson-Pitts, B. J., Huie, R. E., and Orkin, V. L.: Chemical Kinetics and Photochemical Data for Use in Atmospheric Studies, Evaluation Number 15, JPL Publication 06-2, Jet Propulsion Laboratory, Pasadena, CA, 2006.

Schatzmann, M., Olesen, H., Franke, J. (Eds.): COST 732 Model Evaluation Case Studies: Approach and Results, COST Office, ISBN: 3-00-018312-4, 2010.

**Reviewer #2**

Discussion section was more elaborated and improved.

**Question - 1.** Is there a double-counting of the urban processes in the CFD simulations, considering that those simulations are initialized with meteorological fields from HIRLAM model with activated BEP module?

**Answer**: There is no double-counting of urban process, since the BEP module was disabled during operational forecasting with HIRLAM-S03 domain. This fact has been mentioned in the lines 89-91.

**Hence, manuscript text was modified by adding the following sentences**:
**Line 91:** The urban effects were taken into account through roughness parameter and meteorological data assimilated from urban and suburban stations.

**Question - 2.** Are those emissions provided by TNO and used in CAMx simulations the most updated emissions for all the SNAPs?

**Answer**: For this year the EMEP emission inventory was available but at low resolution (0.5 x 0.5 deg.). The emission inventory (at resolution 0.125°×0.0625°) produced by TNO from official country-reported emission data from the year of 2005 was of a better quality. Despite the fact that the emission inventory was 6 years older than the simulations presented in the current research, the inventory had the finest spatial resolution available at that time. Therefore, we favoured TNO emissions since the goal of this study was the high-resolution air-pollution modelling.

**Hence, manuscript text was modified by adding the following sentences**:
**Line 109 (end):** Although, the EMEP emission inventory was also available, but at substantially lower resolution of 0.5 x 0.5 deg., and therefore, the TNO inventory was used.

**Question - 3.** Which kind of k-epsilon turbulence schemes are available in M2EU model? In addition, which schemes are available for meshing?

**Answer**: There are two types of turbulence closure schemes available in M2UE model, i.e. k-eps and cubic eddy-viscosity viscosity scheme by Craft et al., (1996). There is no any specific (advanced) meshing technique available in the M2UE model. An ordinary uniform or non-uniform regular (rectilinear) 3D grid with fictitious domain method (Aloyan et al., 1982; Vabishchevich, 1991) is used in the model.

**Hence, manuscript text was modified by adding the following sentences**:
**Line 199:** equations, k-eps (Launder and Spalding, 1974) and cubic eddy-viscosity (Craft et al., 1996) turbulence closure schemes, and …
**Line 121 after … van Leer, 1974):** "An ordinary uniform or non-uniform regular (rectilinear) 3D grid with fictitious domain method (Aloyan et al., 1982; Vabishchevich, 1991) is used in the model."

**Question - 4.** Authors mentioned that the horizontal and vertical extent of the computational domain is 500 x 500 x 100 m. There is a lack of information about the dimensions of the domain (e.g. height of the tallest building in the domain). Did the authors follow the COST Action 732 guidelines

**Answer**: The M2UE computational domain was constructed following the COST Action 732 best practices and guidelines (see Franke et al., 2007) to ensure minimal influence of inflow/outflow boundaries on the airflow and pollution dispersion. Therefore, for the current micro-scale simulations with a steady-state RANS approach, the distance between buildings and inlet/outlet boundaries was equal to $5H_{max}$, where $H_{max} = 20$ m is the tallest building height. The distance between the tallest building and top boundary was $4H_{max}$. The computational domain could have been larger in extent, in particular, towards the outflow boundary. However, that would not satisfy the operational modelling system requirements because the total runtime interval of the entire modelling chain would exceed 12 hr. Therefore, the current micro-scale domain extent/size is an acceptable compromise between the best practices and guidelines as well as total runtime as requirements.

**Hence, manuscript text was modified by adding the following sentences**:
"The horizontal and vertical extent of the computational domain is 500 x 500 x 100 m (detailed grid is shown in Figure A2). The distance between buildings in the centre of computational domain and inlet/outlet boundaries is $5H_{max}$, where $H_{max} = 20$ m is the tallest building height. The distance between the tallest building and top boundary was $4H_{max}$. Therefore, the current micro-scale domain extent/size is acceptable compromise between the best practices and guidelines (Franke et al., 2007) as well as total operational runtime (should not exceed 12 hr in order to satisfy modelling system requirements for the entire modelling chain)."

**Question - 5.** Authors should provide a figure plotting the mesh to be more specific about this information: the horizontal and vertical numerical grids with local refinement of grid-cells ranging from 1 m at the Jagtvej 130 Street to 5 m near the domains boundaries was used. How about the distance between the boundaries and the build-up area? Again, are you considering the COST Action guidelines or similar?

**Answer**: Detailed M2UE grid is included in Appendix A, Figure A2.

**Hence, manuscript text was modified by adding the following sentences**:
**line 129 after "500 x 500 x 100 m":** "(detailed grid is shown in Figure A2)."

**Question - 6.** The Jagtvej Street emission inventory was derived from hourly traffic counts (Berkowicz et al., 2004) and emissions for different types of road-vehicles (Chao et al., 2018). Therefore, I am assuming you are only considering the road traffic emissions for the M2EU simulations. Am I right?

**Answer**: It is correct, only the most significant pollution source in M2UE computational domain, i.e. traffic pollution, was considered. Road traffic emissions are included in Appendix A, Figure A1.

**Hence, manuscript text was modified by adding the following sentences**:
**Adding to line 131:** "M2UE uses traffic counts measured at the Jagtvej street of Copenhagen to derive more accurate emissions (Figure A1) and hence, account for a very local spatio-temporal features specific for this street."

**Additional references**

Aloyan, A., Baklanov, A., Penenko, V.: Fictitious regions in numerical simulation of quarry ventilation, Soviet Meteorology and Hydrology, 7,32–37, 1982.

Craft, T.J., Launder, B.E., Suga, K.: Development and application of a cubic eddy-viscosity model of turbulence, Int. J. of Heat and Fluid Flow, 17(2), 108-115, https://doi.org/10.1016/0142-727X(95)00079-6, 1996

Franke J., Hellsten, A., Schlünzen, H., Carissimo, B. (Eds.): Best Practice Guideline for the CFD Simulation of Flows in the Urban Environment: COST Action 732 Quality Assurance and Improvement of Microscale Meteorological Models, COST Office, ISBN: 3-00-018312-4, 2007.

Launder, B.E., Spalding, D.B.: "The numerical computation of turbulent flows", Computer Methods in Applied Mechanics and Engineering, 3(2), 269-289, https://doi.org/10.1016/0045-7825(74)90029-2, 1974

Vabishchevich, P.N.: The Method of Fictitious Domains in Problems of Mathematical Physics [in Russian], Moscow University, Moskva, 156pp., 1991.

**Table A1:** List of chemical reactions included as gas-phase chemistry mechanism in M2UE model

| | |
|---|---|
| $NO_2 + hv \longrightarrow O(^3P) + NO$ | $RO_2 + NO \longrightarrow NO_2 + HO_2 + HCHO$ |
| $O_3 + hv \longrightarrow O(^1D) + O_2$ | $CO + HO \longrightarrow HO_2 + CO_2$ |
| $HCHO + hv \longrightarrow 2HO_2 + CO$ | $HO + HC \longrightarrow RO_2 + H_2O$ |
| $HCHO + hv \longrightarrow H_2 + CO$ | $HO + HCHO \longrightarrow HO_2 + CO + H_2O$ |
| $O(^3P) + O_2 \longrightarrow O_3$ | $HO + NO_2 \longrightarrow HNO_3$ |
| $O(^1D) + N_2 \longrightarrow O(^3P) + N_2$ | $2HO_2 \longrightarrow H_2O_2 + O_2$ |
| $O(^1D) + O_2 \longrightarrow O(^3P) + O_2$ | $2HO_2 + H_2O \longrightarrow H_2O_2 + H_2O + O_2$ |
| $O(^1D) + H_2O \longrightarrow 2OH$ | $HO_2 + RO_2 \longrightarrow ROOH + O_2$ |
| $HO_2 + NO \longrightarrow NO_2 + HO$ | $2RO_2 \longrightarrow 2HCHO + HO_2$ |
| $O_3 + NO \longrightarrow NO_2 + O_2$ | $SO_2 + OH \longrightarrow H_2SO_4 + HO_2$ |

Chemical reactions and rate parameters are the same as in Stockwell and Goliff (2002), except chemical reaction ($SO_2 + OH \longrightarrow H_2SO_4 + HO_2$) with the rate parameter from (Sander et al., 2006):

$$k_0 = 3 \times 10^{-31} \times C_{air} \times (300/T)^{3.3}, k_\infty = 1.5 \times 10^{-12}$$

$$k = \frac{k_0}{1 + k_0/k_\infty} \times 0.6^{(1+[\log_{10}(k_0/k_\infty)]^2)^{-1}}$$

where $T$ is temperature [K] and $C_{air}$ is air concentration [molecules/cm³].

[Figure]

**Figure A1:** Time-series of diurnal cycle emissions on typical Saturday, Sunday and Working day of a week at the Jagtvej street, Copenhagen, Denmark.

[Figure]

**Figure A2:** M2UE numerical grid of the Jagtvej street, Copenhagen, Denmark.

---

## Author Comment (AC2)

**Author responses to editor's comments of ACP-2020-1308 - Downscaling system for modelling of atmospheric composition on regional, urban and street scales**

*Roman Nuterman, Alexander Mahura, Alexander Baklanov, Bjarne Amstrup, and Ashraf Zakey*

We are grateful to the editor for critical and valuable comments. Below we provide the response to editor's comments and suggestions.

**Comment 1**. One of the referees is very critical about the publication of your paper. The main critics are listed below. Could you put your work in perspectives to the works of Kwak et al. (2015), Khan et al. (2021), José et al. (2021)?

> **Answer**: Indeed, there are several studies by Kwak et al. (2015), Khan et al. (2021), José et al. (2021), which demonstrated the atmospheric composition modelling with downscaling from regional to street scale. Amongst these only one study (José et al., 2021) used the micro-scale model in an operational mode. But this model has very basic chemistry scheme predominately applicable for cold seasons. The others did not consider an operational aspect and/or unlikely to be able to employ their micro-scale models in operational mode. The operational runtime constraints will not allow doing that because the complexity of gas-phase chemistry mechanism (Kwak et al., 2015) or requirements for numerical grid resolution in large eddy simulation models (Khan et al., 2021).
>
> **Hence, the manuscript text was modified in the following lines**
> Line 48: "Currently, there are several studies describing and evaluating systems capable of downscaling modelling of weather and atmospheric composition from regional to micro-scales. Amongst these the study by José et al. (2021) performed operational micro-scale simulations, but this model has very basic chemistry scheme predominately applicable for cold seasons. The others did not consider an operational aspect and/or unlikely to be able to run their micro-scale models in operational mode. The operational runtime constraints will not allow doing that because the complexity of gas-phase chemistry mechanism (Kwak et al., 2015) or requirements for numerical grid resolution in large eddy simulation models (Khan et al., 2021).
>
> Line 349: Khan, B., Banzhaf, S., Chan, E. C., Forkel, R., Kanani-Sühring, F., Ketelsen, K., Kurppa, M., Maronga, B., Mauder, M., Raasch, S., Russo, E., Schaap, M., and Sühring, M.: Development of an atmospheric chemistry model coupled to the PALM model system 6.0: implementation and first applications, Geosci. Model Dev., 14, 1171-1193, https://doi.org/10.5194/gmd-14-1171-2021, 2021.
>
> Line 355: Kwak, K.-H., Baik, J.-J., Ryu, Y.-H., Lee, S.-H.: Urban air quality simulation in a high-rise building area using a CFD model coupled with mesoscale meteorological and chemistry-transport models, Atmospheric Environment, 100, 167-177, https://doi.org/10.1016/j.atmosenv.2014.10.059, 2015.
>
> Line 398: San José, R., Pérez, J.L., Gonzalez-Barras, R.M.: Assessment of mesoscale and microscale simulations of a NO2 episode supported by traffic modelling at microscopic level, Science of The Total Environment, 752, https://doi.org/10.1016/j.scitotenv.2020.141992, 2021.

**Comment 2**. Furthermore, as the coupling to gas-phase chemistry seems to be an essential part of this work, could you add a comparison to NO2 measurements? The referee suggests that the comparison to NOx only and at only one site is not strong enough to validate the model.

> **Answer**: The comparison to $NO_2$ measurements was added to the manuscript for each used air-quality model in the downscaling chain.

**Hence, the Fig. 4 was replaced by**

[Figure]

Figure 4: Time-series of observations vs. HIRLAM+CAMx+M2UE forecasts for (a) $O_3$, (b, d) $NO_x$ and (c, e) $NO_2$ for the European, Denmark and Copenhagen modelling domains during Sunday-Monday 4-5 Sep 2011; (a, b, c) HCØ roof level observations of $O_3$, $NO_x$ and $NO_2$ vs. modelling results for EU and DK domains; (d, e) Jagtvej Street level observations of $NO_x$ and $NO_2$ vs. M2UE /Pearson correlation $R_p$ and root mean square error $RMSE$ of modelled vs. observed concentrations for 3 domains/.

**and the manuscript text was modified in the following lines**

Line 17: "… forecast of $NO_x$ and $NO_2$ levels …"

Line 21: "… and $NO_2$ diurnal cycles …"

Line 157: "… photochemical reactions of $O_3$, $NO_x$ and $NO_2$ were of …"

Line 158: "… time-series of CFD boundary conditions are identical for $O_3$, $NO_x$ and $NO_2$."

Line 199: "… time-series of $O_3$, $NO_x$ and $NO_2$ observations …"

Line 201: "… only NO$_x$ and NO$_2$, particulate matter …"

Line 205: "Although, the $NO_x$ and $NO_2$ levels were generally overestimated in the DK scale forecast (Fig. 4bc) …"

Line 207: "… ($EU\_RMSE = 23$ vs. $DK\_RMSE = 47$ for $NO_x$ and $EU\_RMSE = 13$ vs. $DK\_RMSE = 33$ for $NO_2$)…"

Line 208: "… ($EU\_R_p = 0.56$ vs. $DK\_R_p = 0.69$ for $NO_x$ and $EU\_R_p = 0.64$ vs. $DK\_R_p = 0.72$ for $NO_2$)."

Line 208: "… when the $NO_x$ and $NO_2$ peaks were observed …"

Line 210: "The timing and levels of $NO_x$ and $NO_2$ peaks were…"

Line 212: "… it with slightly elevated $NO_x$ and $NO_2$ values."

Line 217: "The diurnal cycles of $NO_x$ and $NO_2$ concentrations observed and modelled for the Jagtvej Street are shown in Fig. 4de."

Line 218: "… values ($R_p = 0.66$ and $R_p = 0.45$ for $NO_x$ and $NO_2$, respectively), …"

Line 218: "… observed $NO_x$ and $NO_2$ diurnal cycles …"

Line 221: "… estimate of $NO_x$ concentration …"

Line 224: "Note, the model also exhibited elevated $NO_2$ levels (with RMSE=38) during the high air pollution episode with the concentrations underestimated."

Line 226: "… the elevated $NO_x$ and $NO_2$ concentrations …"

Line 229: "As seen in Fig. 4abc, the effect of …"

Line 260: "… forecast of $NO_x$ and $NO_2$ levels …"

Line 263: "… the $NO_x$ and $NO_2$ diurnal cycles …"

**Comment 3.** The "possibility to run CFD type model operationally coupled with gas-phase chemistry" has been proved by many researches, such as Kwak et. al., (2015), José et. al., (2021) and Khan et. al., (2021). They adopted mesoscale models (e.g., WRF-chem, CMAQ, COSMO) and CFD models (e.g., PALM, MICROSYS) to simulate real world atmospheric environment with gas phase chemistry (including chemistry components such as NO, NO2, O3, CO, …).

**Answer on this comment as on the comment 1**: Indeed, there are several studies by Kwak et al. (2015), Khan et al. (2021), José et al. (2021), which demonstrated the atmospheric composition modelling with downscaling from regional to street scale. Amongst these only one study (José et al., 2021) used the micro-scale model in an operational mode. But this model has very basic chemistry scheme predominately applicable for cold seasons. The others did not consider an operational aspect and/or unlikely to be able to employ their micro-scale models in operational mode. The operational runtime constraints will not allow doing that because the complexity of gas-phase chemistry mechanism (Kwak et al., 2015) or requirements for numerical grid resolution in large eddy simulation models (Khan et al., 2021).

**Comment 4.** Valid modeling settings and reliable evaluation are the basis of operational modeling. In this paper, (1) the boundary conditions are "read at 1-h interval", (2) the total emission values of the whole CFD domain are not equal to the emission values of the corresponding CMAx grid, (3) the horizontal extend is only 500 m, (4) evaluation only conducted on NOx at only one site. Comparatively, the aforementioned studies are more reliable than this work in terms of modeling and evaluation.

**Answer**:

(1) To our knowledge most national meteorological services/institutes run their regional or limited geographical area operational models with output frequency of 1 hour. Since the goal of this study is to demonstrate and test the operational system in the context of current operational practices in the numerical weather prediction and atmospheric composition modelling, the local-scale forecast was also forced with 1-h interval boundary conditions from the meso-scale model.

(2) The anthropogenic emission inventories for most regional-urban air quality models represent annually accumulated fluxes from various sources. Usually, these are redistributed over months, weeks (working days and weekends) and hours with corresponding coefficients. Therefore, these emissions are less accurate than those obtained from local traffic counts. Moreover, the studies by Kwak et al. (2015), Khan

et al. (2021), José et al. (2021) adopted a similar approach to our approach for local scale emissions using either emission models or traffic counts.

(3) The selected street (for M2UE model) domain is rather typical street canyon configuration and a typical section (block) of the lengthy Jagtvej Street. In such configuration the bulk wind direction and speed as well as buildings surrounding the street have the most significant effect on air-flow and traffic pollution dispersion (Karra et al., 2017; Schatzmann et al., 2010).

(4) The comparison to $NO_2$ measurements was added to the manuscript for each used air-quality model in the downscaling chain.

**References**

Karra, S., Malki-Epshtein, L., Neophytou, M. K.-A.: Air flow and pollution in a real, heterogeneous urban street canyon: A field and laboratory study, Atmospheric Environment, Volume 165, 2017, Pages 370-384, https://doi.org/10.1016/j.atmosenv.2017.06.035.

Schatzmann, M., Olesen, H., Franke, J. (Eds.): COST 732 Model Evaluation Case Studies: Approach and Results, COST Office, ISBN: 3-00-018312-4, 2010.